# Assessment of the Technological Properties of Idebenone and Tocopheryl Acetate Co-Loaded Lipid Nanoparticles

**Maria Grazia Sarpietro** , **Cristina Torrisi, Rosario Pignatello** , **Francesco Castelli and Lucia Montenegro ***

Department of Drug and Health Sciences, University of Catania, 95125 Catania, Italy;
mg.sarpietro@unict.it (M.G.S.); torrisi.cristina@hotmail.it (C.T.); rosario.pignatello@unict.it (R.P.);
fcastelli@unict.it (F.C.)
* Correspondence: lmontene@unict.it; Tel.: +39-095-738-4010

**Featured Application: Design and development of nanostructured lipid carriers containing tocopheryl acetate as liquid lipid to obtain a synergic effect with antioxidant drugs loaded into the nanocarriers.**

**Abstract:** Several liquid lipids have been proposed to obtain nanostructured lipid carriers (NLC) with improved efficiency. An attractive strategy is the use of oils that could elicit a synergic effect with the loaded drug. In this work, different percentages (0–4% $w/w$) of tocopheryl acetate (TA), an oily antioxidant, were used as liquid lipid to prepare NLC loading idebenone (IDE), a synthetic antioxidant investigated for the treatment of neurodegenerative and topical diseases. The technological properties of such NLC were evaluated, as well as the interactions among lipid core components. Loading different percentages of IDE (1–4% $w/w$) into NLC containing TA up to 4% $w/w$, no significant change of mean size and polydispersity index was observed. IDE loading capacity was 4% $w/w$ but NLC containing IDE percentages greater than 1.5% $w/w$ showed poor stability during long-term storage. Differential scanning calorimetry analyses highlighted linear relationships between peak temperature and TA percentages, while the enthalpy variation and recrystallization index values showed that increasing the percentage of TA led to less crystalline structure of the NLC core. Therefore, NLC co-loading IDE and TA could be useful to design new delivery systems for the treatment of diseases that could benefit from the co-administration of these antioxidants.

**Keywords:** idebenone; tocopheryl acetate; lipid nanoparticles; differential scanning calorimetry

## 1. Introduction

In the last decades, lipid nanoparticles have gained a great deal of attention in the pharmaceutical field owing to their advantages as drug delivery systems [1–5]. In the 1990s, the first generation of lipid nanoparticles, namely solid lipid nanoparticles (SLN), was developed and investigated as carriers for the delivery of a great variety of active ingredients by different administration routes [6–10]. SLN consist of a solid lipid core stabilized by different types of surfactants in aqueous media. SLN drawbacks, such as poor loading capacity and drug leakage from the nanoparticles during storage, prompted the researchers to design a second generation of lipid nanoparticles, named nanostructured lipid carriers (NLC). The amorphous structure and/or the presence of liquid lipids as oily compartments of the lipid core made these nanoparticles able to accommodate a greater amount of drug compared to SLN, while preventing drug expulsion from the nano-carriers during storage [11,12]. Various solid and liquid lipids have been assessed as components of the NLC core in the attempt to improve the potential therapeutic efficiency of such delivery systems [13]. A promising strategy that could provide NLC with better therapeutic outcomes is the use of liquid lipids with an inherent biological activity that could elicit a synergic effect with the loaded drug. In a previous paper [14], idebenone (IDE), a potent synthetic antioxidant analogous of coenzyme $Q_{10}$ [15,16], was loaded into NLC

whose lipid components were cetyl palmitate, a solid wax of GRAS (Generally Regarded As Safe) status, and tocopheryl acetate (TA), a lipophilic liquid that is regarded as a potent antioxidant in the human body [17,18]. After topical treatment of human volunteers with a gel containing NLC co-loading IDE and TA, an increased photo-protective activity was observed compared to IDE-loaded SLN incorporated in the same vehicle. However, in the abovementioned study, the effects of co-loading different percentages of IDE and TA on the technological properties of the resulting NLC were not investigated. As the knowledge of the most suitable ratio between IDE and TA could be helpful in designing NLC with improved biological activity, in this work, NLC containing simultaneously different concentrations of IDE and TA were prepared and the influence of co-loading these antioxidants on lipid nanoparticles features such as mean sizes, polydispersity index, $\zeta$-potential, loading capacity, and long-term stability was assessed. The colloidal suspensions that proved to be more stable were analyzed by differential scanning calorimetry (DSC) to elucidate the key parameters involved in TA and IDE interactions. The results of this study pointed out that an increase of TA content improved IDE loading capacity of the resulting NLC but led to colloidal suspensions with limited stability at room temperature. TA could be incorporated into the NLC core up to 3.5% $w/w$ without significantly affecting the technological properties of the nano-carriers, allowing loading simultaneously IDE percentages as high as 1.5% $w/w$. DSC analyses revealed that increasing the amount of TA and IDE co-loaded into NLC resulted in a decrease of both the enthalpy variation and recrystallization index owing to the formation of a more amorphous lipid core that could allow for designing more efficient controlled-release formulations.

Therefore, the development of lipid nanoparticles co-loading IDE and TA could provide a promising tool for the design of new delivery systems for the treatment of neurodegenerative and topical diseases that benefit from the simultaneous administration of these two antioxidants.

## 2. Materials and Methods

### 2.1. Materials

Idebenone (IDE) was bought from Carbosynth (Berkshire, UK). Polyoxyethylene-20-oleyl ether (Brij 98®, Oleth-20) and tocopheryl acetate (TA) were obtained from Farmalabor (Canosa di Puglia, Italy). Glyceryl oleate (Tegin O®, GO), cetyl palmitate (CP), and imidazolidinyl urea (Kemipur 100®) were purchased from ACEF (Fiorenzuola D'Arda, Italy). All other reagents were of analytical grade.

### 2.2. Preparation of Lipid Nanoparticles

The composition of lipid nanoparticles (SLN and NLC) containing different percentages of IDE and TA is illustrated in Table 1. All nanoparticles were prepared using the phase inversion temperature (PIT) method, as previously reported [19]. Briefly, after heating at 90 °C the oil phase components (reported in Table 1) and the aqueous phase separately, the aqueous phase was slowly added to the oil phase under stirring (700 rpm). The aqueous phase consisted of deionized water containing imidazolidinyl urea as preservative (0.35% $w/w$). The resulting colloidal suspension was cooled down to room temperature, and at the phase inversion temperature (PIT), the turbid mixture turned into clear. The PIT value was recorded using a conductivity meter (model 525, Crison, Modena, Italy). Lipid nanoparticle samples were stored in airtight vials at room temperature and sheltered from the light until used.

### 2.3. Transmission Electron Microscopy (TEM)

Images of unloaded and IDE-loaded lipid nanoparticles were taken using a transmission electron microscope (model JEM 2010, Jeol, Peabody, MA, USA) operating at an acceleration voltage of 200 KV. Imaging was preceded by negative-staining, placing 5 µL of undiluted sample on a 200-mesh formvar copper grid (TAAB Laboratories Equipment, Berks, UK) and allowing it to be adsorbed. The sample surplus was removed using filter

paper, and then a drop of 2% (*w/v*) aqueous solution of uranyl acetate was added over 2 min. The samples were analyzed after drying at room temperature.

**Table 1.** Composition (% *w/w*) of lipid nanoparticles containing different percentages of tocopheryl acetate and idebenone. GO = glyceryl oleate; CP = cetyl palmitate; TA = tocopheryl acetate, IDE = idebenone.

| Code | Oleth-20 | GO | CP | TA | IDE |
|---|---|---|---|---|---|
| SLN | 8.7 | 4.4 | 7.0 | - | - |
| VIT 1 | 8.7 | 4.4 | 6.0 | 1.0 | - |
| VIT 2 | 8.7 | 4.4 | 5.0 | 2.0 | - |
| VIT 3 | 8.7 | 4.4 | 4.0 | 3.0 | - |
| VIT 3,5 | 8.7 | 4.4 | 3.5 | 3.5 | - |
| VIT 4 | 8.7 | 4.4 | 3.0 | 4.0 | - |
| SLN IDE1 | 8.7 | 4.4 | 7.0 | - | 1.0 |
| VIT 1 IDE1 | 8.7 | 4.4 | 6.0 | 1.0 | 1.0 |
| VIT 1 IDE1,5 | 8.7 | 4.4 | 6.0 | 1.0 | 1.5 |
| VIT 1 IDE2 | 8.7 | 4.4 | 6.0 | 1.0 | 2.0 |
| VIT 2 IDE1 | 8.7 | 4.4 | 5.0 | 2.0 | 1.0 |
| VIT 2 IDE 1,5 | 8.7 | 4.4 | 5.0 | 2.0 | 1.5 |
| VIT 2 IDE2 | 8.7 | 4.4 | 5.0 | 2.0 | 2.0 |
| VIT 3 IDE1 | 8.7 | 4.4 | 4.0 | 3.0 | 1.0 |
| VIT 3 IDE1,5 | 8.7 | 4.4 | 4.0 | 3.0 | 1.5 |
| VIT 3 IDE2 | 8.7 | 4.4 | 4.0 | 3.0 | 2.0 |
| VIT 3 IDE2,5 | 8.7 | 4.4 | 4.0 | 3.0 | 2.5 |
| VIT 3 IDE3 | 8.7 | 4.4 | 4.0 | 3.0 | 3.0 |
| VIT 3 IDE3,5 | 8.7 | 4.4 | 4.0 | 3.0 | 3.5 |
| VIT 3,5 IDE1 | 8.7 | 4.4 | 3.5 | 3.5 | 1.0 |
| VIT 3,5 IDE1,5 | 8.7 | 4.4 | 3.5 | 3.5 | 1.5 |
| VIT 3,5 IDE2 | 8.7 | 4.4 | 3.5 | 3.5 | 2.0 |
| VIT 3,5 IDE2,5 | 8.7 | 4.4 | 3.5 | 3.5 | 2.5 |
| VIT 3,5 IDE3 | 8.7 | 4.4 | 3.5 | 3.5 | 3.0 |
| VIT 3,5 IDE3,5 | 8.7 | 4.4 | 3.5 | 3.5 | 3.5 |
| VIT 3,5 IDE4 | 8.7 | 4.4 | 3.5 | 3.5 | 4.0 |
| VIT 4 IDE1 | 8.7 | 4.4 | 3.0 | 4.0 | 1.0 |
| VIT 4 IDE1,5 | 8.7 | 4.4 | 3.0 | 4.0 | 1.5 |
| VIT 4 IDE2 | 8.7 | 4.4 | 3.0 | 4.0 | 2.0 |

*2.4. Photon Correlation Spectroscopy (PCS)*

Lipid nanoparticle sizes and polydispersity indexes (PDI) were determined by photon correlation spectroscopy (PCS), while zeta potential ($\zeta$-potential) values were obtained by Laser Doppler Velocimetry (LDV). All measurements were performed using a Zetasizer Nano ZS90 (Malvern Instruments, Malvern, UK). PCS analyses were carried out by scattering light at 90° using a 4 mW laser diode operating at 670 nm. Samples were diluted 1:4 *v/v* with filtered deionized water prior to measurements. For zeta potential determination, samples were diluted (1:1) with KCl 1 mM (pH 7.0) and analyzed according to a procedure previously reported [20]. All measurements were performed at room temperature. Mean diameter, PDI, and zeta potential values were calculated as the average of results obtained for three replicates of two separate preparations.

*2.5. Stability Tests*

Samples of lipid nanoparticles were stored for 12 months at room temperature, sheltered from the light. Particle size, PDI, and $\zeta$-potential were assessed monthly during the first three months and after 6 and 12 months by PCS or LDV, as described in Section 2.4.

*2.6. Differential Scanning Calorimetry Analyses*

Calorimetric analyses were performed by a Mettler Toledo STARe system (Switzerland) equipped with a DSC-822$^e$ calorimetric cell; a Mettler TA-STAR$^e$ software was used. The

sensitivity was automatically chosen as the maximum possible by the calorimetric system. The reference pan was filled with the same vehicle of the samples under study. The calorimetric system was calibrated, in temperature and enthalpy changes, following the procedure of the DSC 822 Mettler TA STAR$^e$ instrument, by using indium, stearic acid, and cyclohexane as standards. In total, 100 µL of sample was put into the calorimetric pan, hermetically sealed and submitted to analysis as follows: (i) a heating scan from 5 to 65 °C (2 °C/min); (ii) a cooling scan from 65 to 5 °C (4 °C/min) at least three times. Each analysis was carried out in triplicate. The melting enthalpy (ΔH) was obtained by integration of the area under the transition peak. The recrystallization index (RI), expressed as a percentage, was calculated from the following equation [21]:

$$RI(\%) = [\Delta H_{\text{aq. lipid np disp.}} / (\Delta H_{\text{bulk material}} \times \text{conc}_{\text{sol. lip. phase}})] \times 100$$

where $\Delta H_{\text{aq. lipid np disp.}}$ is ΔH of the aqueous lipid nanoparticle dispersion, $\Delta H_{\text{bulk material}}$ is calculated as cetyl palmitate enthalpy variation and $\text{conc}_{\text{sol. lip. phase}}$ is the concentration of the solid lipid phase.

## 3. Results and Discussion

Unloaded lipid nanoparticles showed mean sizes in the range 26–38 nm (Table 2). The comparison between the mean size of unloaded SLN and unloaded NLC pointed out that the inclusion in the nanoparticle lipid core of percentages of TA up to 3.5% *w/w* decreased the nanoparticle mean size, while TA percentages as high as 4% *w/w* led to an increase of nanoparticle mean sizes. However, no clear relationship between TA percentage and nanoparticle mean size could be outlined. PDI values for unloaded SLN and unloaded NLC containing TA percentages up to 3.5% *w/w* were similar and lower than 0.300, thus suggesting that monodisperse systems were obtained. On the contrary, unloaded NLC containing TA 4% *w/w* showed a much higher PDI value (0.55), owing to a non-homogenous dimensional distribution. All unloaded SLN and NLC had similar and slightly negative zeta potential values, regardless of the percentage of TA incorporated in the lipid core.

As reported in literature [22], when emulsified systems are involved, the higher the PIT value, the greater the system stability. Therefore, formulation VIT 3 could be expected to show a greater stability than unloaded SLN and other unloaded NLC. The results of stability studies performed storing at room temperature unloaded SLN and NLC up to 12 months are depicted in Figure 1. Apart from formulation VIT 4, no significant alteration of nanoparticle mean size was observed. During storage, zeta potential values remained unchanged for all unloaded lipid nanoparticles, while PDI slightly increased only for formulation VIT 4 (data not shown). As all unloaded SLN and NLC showed similar zeta potential values, the lower stability of formulation VIT 4 could be attributed to its higher PDI value due to a non-homogenous size distribution that could more easily lead to nanoparticles aggregation during storage.

These results suggest that the incorporation into the nanoparticle lipid core of TA as liquid lipid strongly affected the resulting technological properties such as mean size, PDI, and PIT, depending on the percentage of TA used. However, no significant morphological change was observed due to the incorporation of TA into the lipid nanoparticles, as shown in Figure 2. Regardless of the percentage of TA in the colloidal suspension, nanoparticles were approximately round-shaped with no sign of aggregation. As similar images were obtained for all lipid nanoparticles (unloaded and IDE-loaded SLN and NLC), in Figure 2, only two examples of unloaded lipid nanoparticles were illustrated.

**Table 2.** Mean size, polydispersity index (PDI), ζ-potential (Zeta) and phase inversion temperature (PIT) of unloaded and IDE-loaded lipid nanoparticles.

| Sample | Size ± S.D. (nm) | PDI ± S.D. | Zeta ± S.D. (mV) | PIT (°C) |
|---|---|---|---|---|
| SLN | 35.5 ± 0.9 | 0.22 ± 0.01 | −7.52 | 78 |
| VIT 1 | 28.8 ± 0.9 | 0.23 ± 0.03 | −6.89 | 85 |
| VIT 2 | 28.0 ± 0.6 | 0.23 ± 0.05 | −7.34 | 78 |
| VIT 3 | 26.0 ± 0.5 | 0.18 ± 0.01 | −8.14 | 82 |
| VIT 3,5 | 26.8 ± 0.8 | 0.22 ± 0.01 | −7.66 | 85 |
| VIT 4 | 38.6 ± 0.4 | 0.55 ± 0.02 | −8.39 | 72 |
| SLN IDE1 | 36.4 ± 0.3 | 0.26 ± 0.01 | −8.43 | 78 |
| VIT 1 IDE1 | 26.8 ± 1.3 | 0.18 ± 0.04 | −7.91 | 78 |
| VIT 1 IDE1,5 | 24.1 ± 0.3 | 0.16 ± 0.04 | −6.54 | 65 |
| VIT 1 IDE2 | N.D. [a] | N.D. [a] | N.D. [a] | 62 |
| VIT 2 IDE1 | 25.3 ± 0.6 | 0.17 ± 0.03 | −7.01 | 75 |
| VIT 2 IDE 1,5 | 23.2 ± 0.1 | 0.11 ± 0.01 | −7.55 | 68 |
| VIT 2 IDE2 | N.D. [a] | N.D. [a] | N.D. [a] | 63 |
| VIT 3 IDE1 | 25.3 ± 0.8 | 0.16 ± 0.02 | −8.69 | 75 |
| VIT 3 IDE1,5 | 23.3 ± 0.2 | 0.15 ± 0.01 | −7.31 | 69 |
| VIT 3 IDE2 | 23.6 ± 0.5 | 0.19 ± 0.08 | −8.33 | 67 |
| VIT 3 IDE2,5 | 24.5 ± 0.3 | 0.24 ± 0.01 | −7.49 | 62 |
| VIT 3 IDE3 | 27.8 ± 0.5 | 0.20 ± 0.01 | −7.78 | 60 |
| VIT 3 IDE3,5 | N.D. [a] | N.D. [a] | N.D. [a] | 57 |
| VIT 3,5 IDE1 | 25.3 ± 0.2 | 0.17 ± 0.01 | −6.98 | 70 |
| VIT 3,5 IDE1,5 | 22.1 ± 0.1 | 0.16 ± 0.04 | −6.71 | 66 |
| VIT 3,5 IDE2 | 23.5 ± 0.4 | 0.18 ± 0.06 | −7.86 | 64 |
| VIT 3,5 IDE2,5 | 25.4 ± 0.3 | 0.19 ± 0.03 | −8.76 | 63 |
| VIT 3,5 IDE3 | 28.8 ± 0.2 | 0.18 ± 0.01 | −8.43 | 62 |
| VIT 3,5 IDE3,5 | 32.4 ± 0.4 | 0.16 ± 0.01 | −7.53 | 60 |
| VIT 3,5 IDE4 | 36.7 ± 0.5 | 0.15 ± 0.01 | −6.99 | 59 |
| VIT 4 IDE1 | 25.7 ± 1.5 | 0.13 ± 0.03 | −8.43 | 70 |
| VIT 4 IDE1,5 | 22.2 ± 0.2 | 0.12 ± 0.01 | −7.42 | 64 |
| VIT 4 IDE2 | N.D. [a] | N.D. [a] | N.D. [a] | 58 |

[a] N.D. = not determined, as a slight precipitate was observed after 24 h.

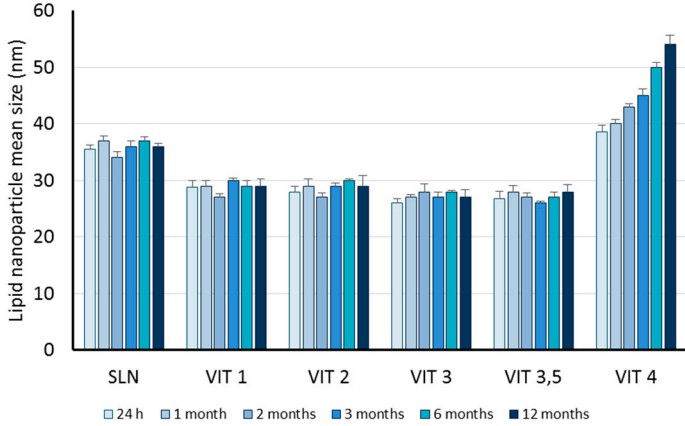

**Figure 1.** Mean sizes of unloaded solid lipid nanoparticles (SLN) and nanostructured lipid carriers (NLC) during storage at room temperature for 12 months.

As shown in Table 2, IDE loading capacity of the nanoparticles under investigation depended on the percentage of TA incorporated into the lipid core. According to previous studies [14], the loading capacity was determined as the maximum amount of IDE that could be incorporated in the colloidal system without any sign of precipitation. Jenning et al. [23] claimed that when water-insoluble compounds are incorporated into lipid nanoparticles, if the resulting colloidal dispersion is clear, all drugs must be in the lipid

phase of the colloidal dispersion. Therefore, in clear colloidal dispersion, the incorporation rate of a lipophilic drug is for all practical purposes 100%. In a previous work [20], we determined that IDE water solubility was 5 µg/mL. Therefore, when we obtained clear lipid nanoparticles dispersions, we considered that all added drug was entrapped into the nanoparticle lipid core. IDE loading capacity was 1% *w/w* for SLN, and it increased by introducing TA up to 3.5% *w/w* into the nanoparticle lipid core (IDE loading capacity: 1.5% *w/w* for VIT 1 and VIT 2; 3% *w/w* for VIT 3; 4% *w/w* for VIT 3,5). Unexpectedly, although formulation VIT 4 contained the highest percentage of TA, its loading capacity was similar to that of VIT 1 and VIT 2, thus suggesting the existence of an optimal percentage of liquid lipid to obtain the maximum drug loading.

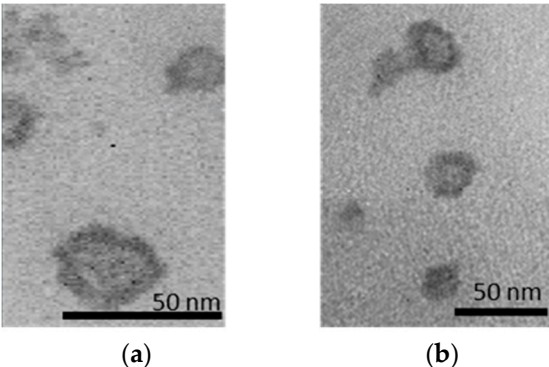

(a)          (b)

**Figure 2.** Transmission electron microscopy (TEM) images of (**a**) unloaded SLN and (**b**) unloaded NLC containing tocopheryl acetate 1% *w/w*.

IDE-loaded lipid nanoparticles showed mean sizes in the range 22–37 nm, narrow dimensional distributions (PDI < 0.300), and slightly negative zeta potentials. It is interesting to note that an increase of nanoparticle mean size was observed for formulations containing TA 3.5% *w/w* owing to the incorporation of greater amount of IDE, although no linear relationship could be pointed out. As illustrated in Figure 3, a parabolic relationship was observed between the percentages of TA used to prepare NLC loaded with IDE 1.5% *w/w* and the PIT values of the resulting nanoparticles, and the maximum PIT value was observed for nanoparticles containing TA 3% *w/w* (formulation VIT 3 IDE 1,5). Conversely, by plotting PIT values against the amount of IDE-loaded into formulations containing TA 3.0 or 3.5% *w/w*, an almost linear decrease of PIT values was observed by increasing the incorporated amount of IDE (Figure 4; $r^2$ = 0.969 for VIT3; $r^2$ = 0.942 for VIT 3,5). These results suggest that formulations containing greater amounts of IDE and TA could be less stable because of their lower PIT values.

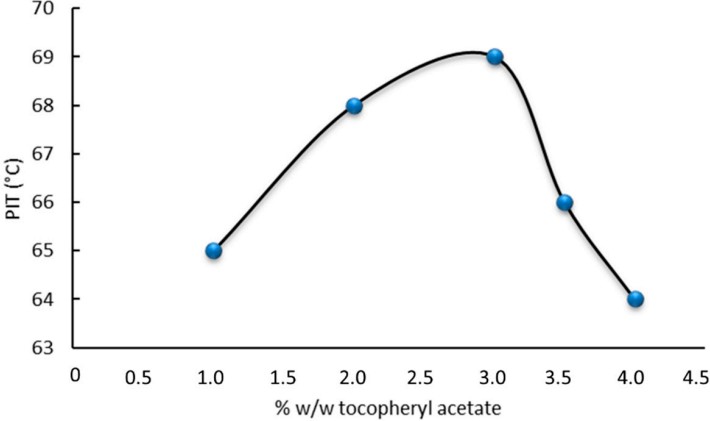

**Figure 3.** Relationship between PIT values and% *w/w* of tocopheryl acetate of NLC loaded with IDE 1.5% *w/w*.

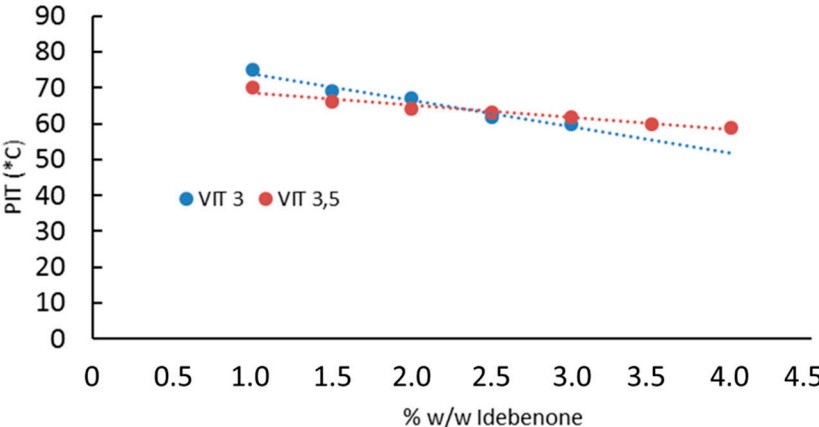

**Figure 4.** Relationship between PIT values and percentages of idebenone of NLC containing tocopheryl acetate 3.0% *w/w* (VIT 3) or tocopheryl acetate 3.5% *w/w* (VIT 3,5).

Stability studies performed storing IDE-loaded lipid nanoparticles for 12 months at room temperature (sheltered from the light) did not highlight any significant change of nanoparticle mean size for formulations containing IDE 1 and 1.5% *w/w*, regardless of the percentage of TA used to prepare the colloidal dispersion (Figure 5a). The incorporation of a greater amount of IDE led to lower stability of the resulting nanoparticles. In particular, formulations containing IDE percentages greater than 1.5% *w/w* gave rise to a slight precipitate after 1–3 months of storage, depending on the TA content (see Figure 5b,c). PDI and ζ-potential values did not significantly change for all investigated lipid nanoparticles (data not shown).

To give insight into the interactions among IDE and the components of the nanoparticle lipid core, DSC studies were performed on the most stable lipid nanoparticles. Unloaded SLN and NLC were used as a reference to investigate the thermal behavior of IDE-loaded SLN and NLC with long-term stability loading of the highest percentage of IDE 1.5% *w/w* (formulations VIT 1 IDE1,5, VIT 2 IDE1,5, VIT 3 IDE 1,5, VIT 3,5 IDE 1,5 and VIT 4 IDE 1,5).

In Figure 6, calorimetric curves of SLN and NLC prepared with increasing amount of TA are shown. The calorimetric curve of SLN was characterized by a main peak at 41.90 °C and a shoulder at a lower temperature. The shoulder could indicate a non-homogenous distribution of the surfactant in the SLN structure. The presence of TA in the nanoparticles produced large variations in the calorimetric curve. NLC VIT 1 showed a main peak at about 40.00 °C and a shoulder at lower temperature. A single peak was present in the calorimetric curve of NLC VIT 2 but at lower temperature compared to the main peak of NLC VIT 1. Increasing the amount of TA resulted in the shift of the main peak to lower temperatures and in the decrease of its entity up to TA 3.5% *w/w*, while the calorimetric curve of NLC VIT 4 was an almost flat line, thus suggesting that the lipid core was no longer in the solid state.

Calorimetric curves of NLC containing IDE are shown in Figure 7 and compared with those of unloaded and IDE-loaded SLN. The calorimetric curve of SLN IDE showed a main peak at about 41.39 °C and a shoulder at a higher temperature. These two signals were previously attributed to a prevalent location of IDE at the lipid/surfactant interface [24]. In VIT1 IDE, a main peak at about 40.30 °C and a shoulder at lower temperature were observed. Increasing the amount of TA caused the shoulder to disappear, whereas the main peak moved to a lower temperature, becoming smaller and smaller until it disappeared almost completely in VIT 3,5 IDE 1,5.

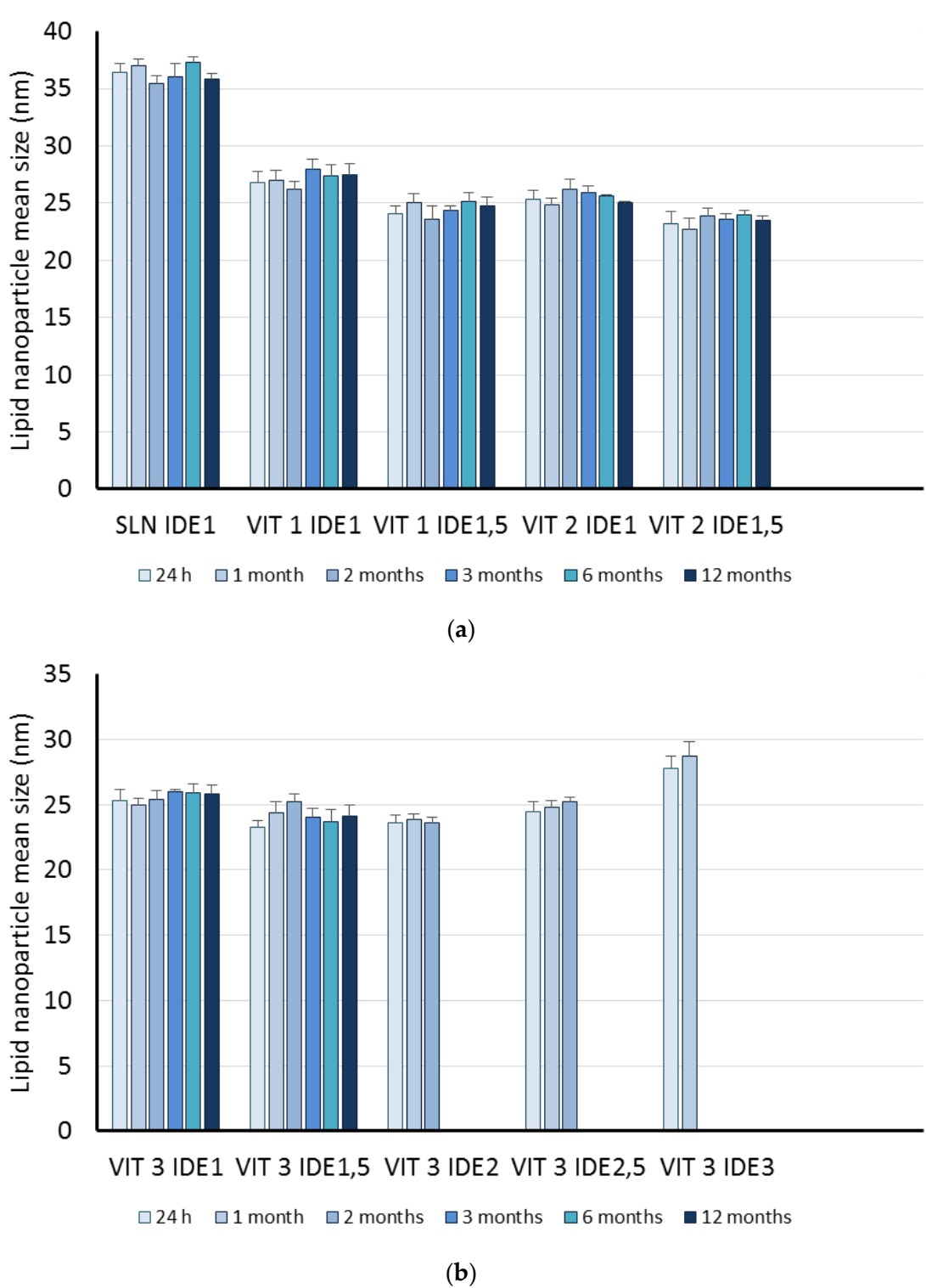

(**a**)

(**b**)

**Figure 5.** *Cont.*

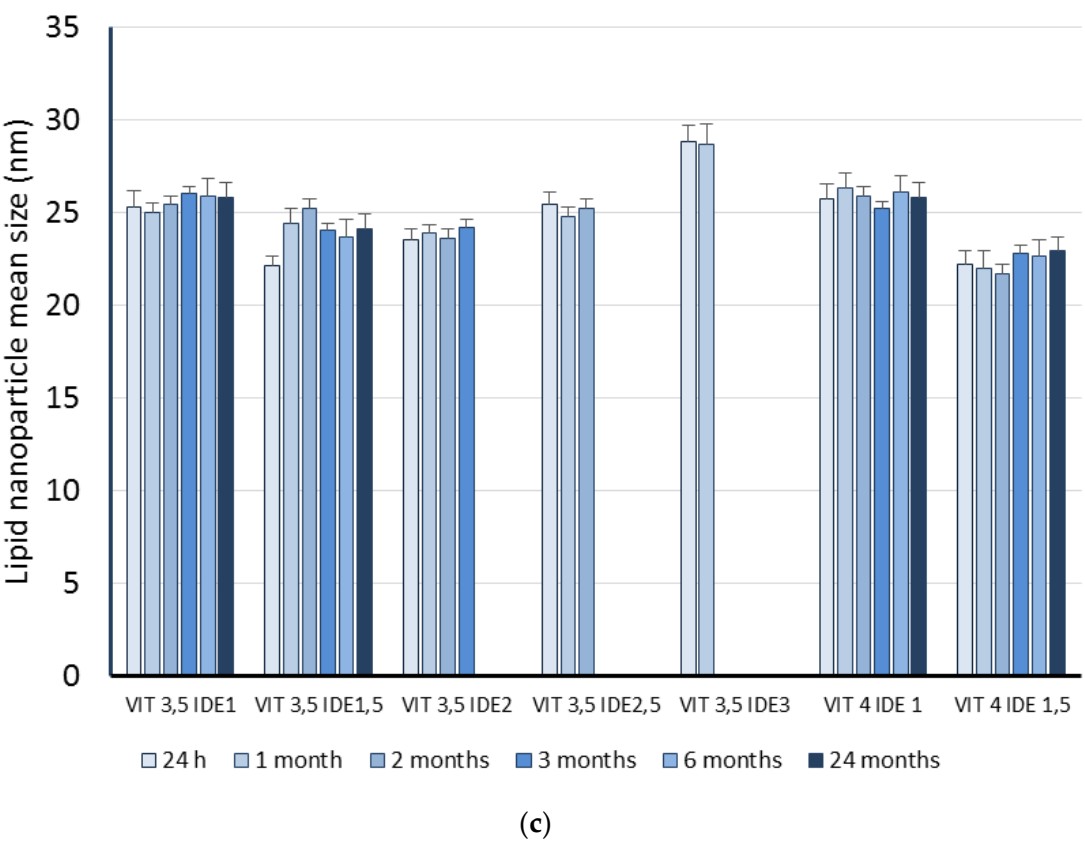

**(c)**

**Figure 5.** Mean sizes of idebenone (IDE)-loaded SLN and NLC during storage at room temperature for 12 months. (**a**) SLN loaded with IDE 1% *w/w* and NLC containing tocopheryl acetate 1% or 2% *w/w* and different percentages of IDE (VIT 1 IDE1, VIT 1 IDE1,5, VIT 2 IDE1, VIT 2 IDE1,5); (**b**) NLC containing tocopheryl acetate 3% and different percentages of IDE (VIT 3 IDE1, VIT 3 IDE 1,5, VIT 3 IDE 2, VIT 3 IDE 2,5, VIT 3 IDE 3); (**c**) NLC containing tocopheryl acetate 3.5% or 4% *w/w* and different percentages of IDE. No bars are shown in the plot when nanoparticle mean size was not determined, owing to the presence of a precipitate in the colloidal suspension. Each stability test was performed in triplicate.

In NLC, regardless of the presence of IDE, at the lowest amount of TA a well evident shoulder was present, suggesting a non-homogeneous distribution of the components in the NLC structure. For a greater amount of TA, only one peak was visible, indicating a homogeneous NLC structure.

Plots of the peak temperature and the enthalpy variation (ΔH) values vs. TA percentage in unloaded and IDE-loaded lipid nanoparticles are reported in Figures 8 and 9, respectively. The addition of TA to SLN to produce NLC caused the reduction of the transition temperature as well as of the enthalpy variation. In addition, increasing the amount of TA both in unloaded and IDE-loaded NLC led to the decrease of the transition temperature and of the enthalpy variation. Linear relationships were observed between peak temperature decrease and TA percentage in unloaded and IDE-loaded lipid nanoparticles ($r^2 = 0.98$ for unloaded lipid nanoparticles; $r^2 = 0.97$ for IDE-loaded lipid nanoparticles). A similar trend was observed for plotting ΔH values against TA percentage in unloaded and IDE-loaded lipid nanoparticles ($r^2 = 0.99$ for unloaded lipid nanoparticles; $r^2 = 0.98$ for IDE-loaded lipid nanoparticles).

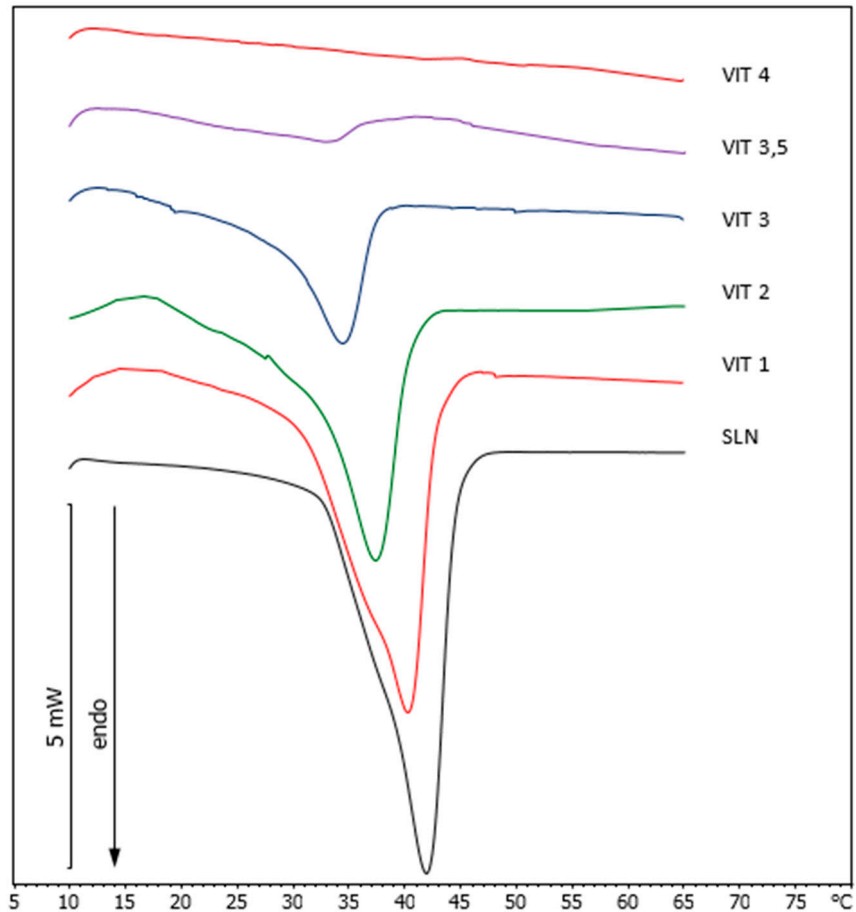

**Figure 6.** Differential scanning calorimetry (DSC) curves of unloaded SLN and NLC.

From the comparison of the NLC without and with IDE, interesting information can be obtained. With regard to the peak temperature, it can be noticed that this parameter decreased by increasing the amount of TA, while only slight differences were observed between NLC without and with IDE. This behavior suggests that TA caused a destabilization of the nanoparticles structure, while the presence of IDE did not seem to affect the stability of the NLC. On the other hand, the enthalpy variation decrease of the NLC was more pronounced in the presence of IDE. The enthalpy variation decrease is an indication of a lower cooperativity among the lipid molecules during the transition [25]. Therefore, it is possible to hypothesize that IDE did not alter the stability of the system but decreased the cooperativity of the lipids.

According to do Prado et al. [26], less crystalline structures require less energy to promote the fusion of the compound; thus, lower enthalpy values are expected. Therefore, the reduction of ΔH due to TA incorporation into the nanoparticles suggest that this lipophilic liquid contributed to the formation of a more amorphous structure. This hypothesis is supported by the lower recrystallization index observed for colloidal suspensions containing TA (Table 3).

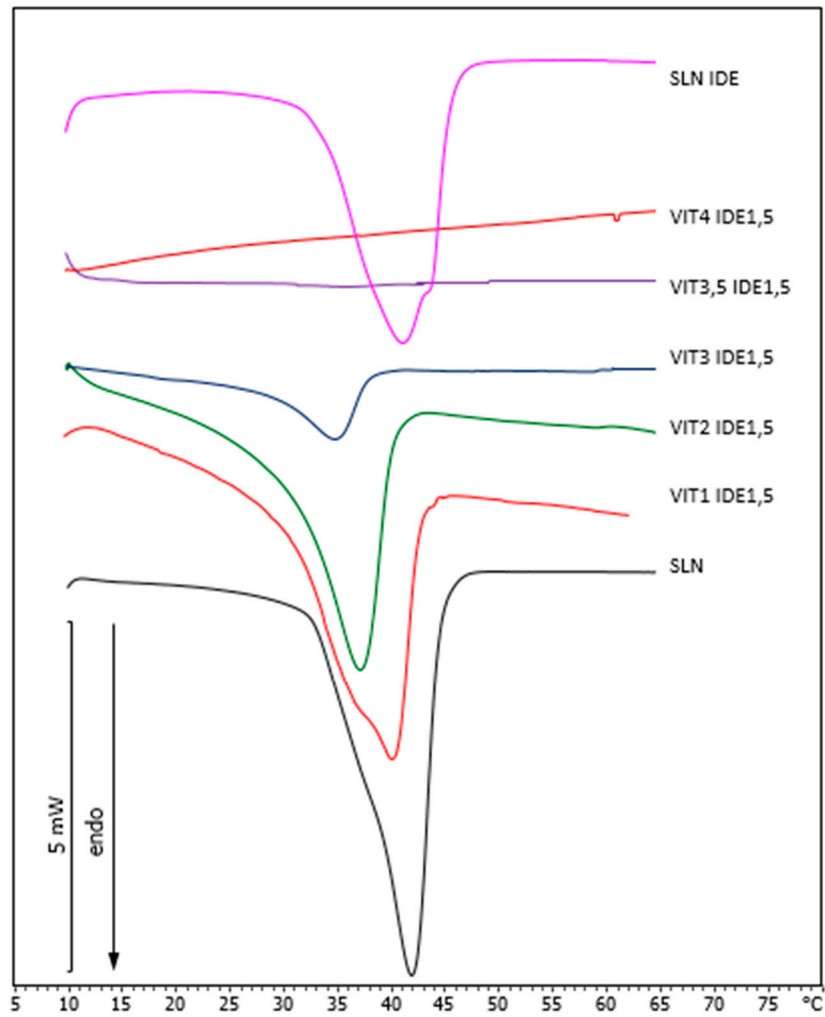

**Figure 7.** DSC curves of unloaded and IDE-loaded SLN, NLC loaded with IDE 1.5% *w/w*.

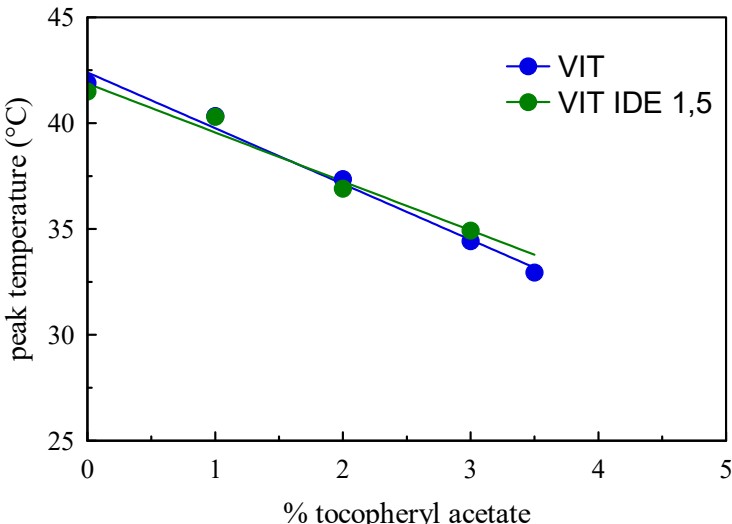

**Figure 8.** Peak temperature of NLC (unloaded NLC = VIT; 1.5% *w/w*, IDE-loaded NLC = VIT IDE1,5), as a function of the percentage of tocopheryl acetate.

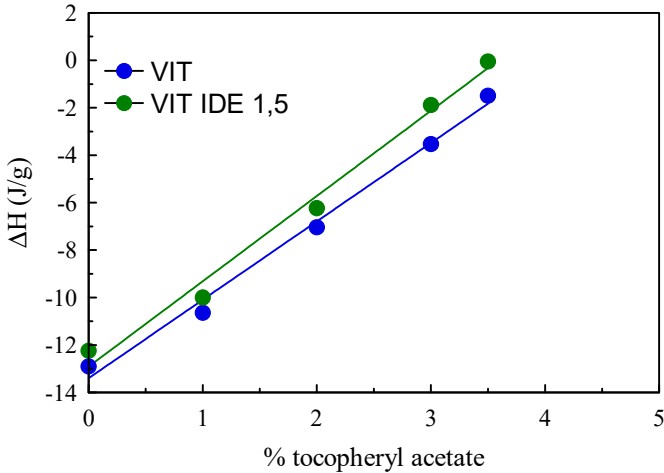

**Figure 9.** Enthalpy variation (ΔH) of NLC (unloaded NLC = VIT; 1.5% *w/w*, IDE-loaded NLC = VIT IDE1,5), as a function of the percentage of tocopheryl acetate.

**Table 3.** Enthalpy changes (ΔH) and recrystallization index (RI%) of unloaded SLN and NLC, IDE-loaded SLN, and 5% *w/w* IDE-loaded NLC.

| Sample | ΔH (J/g) | RI (%) |
| --- | --- | --- |
| Cetyl palmitate | −247.00 | 100.00 |
| SLN | −12.90 | 74.61 |
| VIT 1 | −10.64 | 71.79 |
| VIT 2 | −7.04 | 5.00 |
| VIT 3 | −3.53 | 35.72 |
| VIT 3,5 | −1.49 | 17.23 |
| VIT 4 | - | - |
| SLN IDE | 12.24 | 70.79 |
| VIT 1 IDE1,5 | 10.00 | 67.47 |
| VIT 2 IDE1,5 | 6.23 | 50.44 |
| VIT 3 IDE1,5 | 1.88 | 19.02 |
| VIT 3,5 IDE1,5 | 0.05 | 0.58 |
| VIT 4 IDE1,5 | - | - |

As reported in literature [27], the lipid crystalline structure is a key factor in determining whether a drug is expelled or firmly incorporated into the carrier. In particular, the decrease in crystallinity is favorable for compounds entrapment, as lattice defects of the lipid structure could offer space to accommodate the drugs, while high crystalline lipids would lead to drug expulsion [27]. Therefore, the less ordered crystalline structure conferred by IDE could prevent the undesired and premature release of the incorporated drug from the nanoparticles, leading to more stable colloidal suspensions and allowing the modulation of drug release depending on the liquid lipid content.

IDE loading into lipid nanocarriers has been explored as a strategy to improve both topical and brain delivery of this synthetic antioxidant [16,28,29]. Recently, IDE-loaded NLC prepared using oleic acid and cetyl palmitate as liquid and solid lipid, respectively, have been investigated for the treatment of mitochondrial dysfunctions involved in several neurodegenerative diseases [30], providing promising results for the development of future formulations. The use of tocopheryl acetate as a liquid lipid to prepare IDE-loaded NLC could be regarded as an advantageous strategy as it allows for obtaining lipid nanoparticles whose technological properties could be modulated depending on TA content while incorporating a liquid lipid that is considered beneficial in the treatment of both neurological [31] and topical disorders [32].

Therefore, further studies have been planned to investigate the effectiveness of IDE and TA co-loaded lipid nanoparticles in in vitro models of neurodegenerative and skin diseases.

**Author Contributions:** Conceptualization, L.M.; methodology, L.M. and M.G.S.; validation, L.M. and M.G.S.; formal analysis, L.M. and M.G.S.; investigation, L.M., M.G.S., C.T.; resources, L.M., M.G.S., R.P., F.C.; writing—original draft preparation, L.M. and M.G.S.; writing—review and editing, L.M., M.G.S., R.P., F.C.; visualization, L.M. and M.G.S.; supervision, L.M.; project administration, L.M.; funding acquisition, L.M. All authors have read and agreed to the published version of the manuscript.

**Funding:** This research was funded by the University of Catania, Bando FIR 2014, Progetto di ricerca Cod. 764AAD.

**Institutional Review Board Statement:** Not applicable.

**Informed Consent Statement:** Not applicable.

**Data Availability Statement:** Data is contained within the article.

**Conflicts of Interest:** The authors declare no conflict of interest.

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
