# Peer review of "Assessment of the Technological Properties of Idebenone and Tocopheryl Acetate Co-Loaded Lipid Nanoparticles"

_applsci, doi:10.3390/app11083553_

Round 1

Reviewer 1 Report

The authors have provide detailed results to support their claim of the NLC delivery system. 

Did the authors detected leak of IDE from the loaded particles over time?

What is the mechanism/trigger of the release of IDE from the loaded particles?

How did this new NLC compares to the traditional iron-oxide based NLCs?

Author Response

Reviewer 1

The authors have provide detailed results to support their claim of the NLC delivery system. 

We would like to thank the reviewer for reviewing our manuscript and for his/her comments.

  1. Did the authors detected leak of IDE from the loaded particles over time?

Answer

Owing to IDE poor water solubility, IDE leakage from the nanoparticles would lead to a precipitate. As reported in the manuscript (page 7 and Fig. 5 a, b, c), during storage of lipid nanoparticles loading IDE percentage smaller than 1.5% w/w we did not observe any significant change. As these formulations were clear, no leakage of IDE could have occurred. During storage of lipid nanoparticles loading IDE percentage greater than 1.5% w/w, we detected a slight precipitate depending on idebenone and tocopheryl acetate concentration. This precipitate looked like a flocculate and IDE leakage could have occurred. We did not perform any analysis to determine the presence of free IDE in this precipitate because the formation of even a slight precipitate, regardless of its composition, was a sign of instability of the colloidal dispersion.  

  1. What is the mechanism/trigger of the release of IDE from the loaded particles?

Answer

As reported in literature (zur Mühlen et al., Solid lipid nanoparticles (SLN) for controlled drug delivery– Drug release and release mechanism. Eur. J. Pharm. Biopharm. 1998, 45, 149-155), the main mechanism involved in drug release from lipid nanoparticles is drug diffusion through the lipid core. This mechanism is affected by several factors, including drug lipid interactions, nanoparticle size, matrix viscosity, and length of the diffusion pathway. We have planned to investigate IDE release from lipid nanoparticles in a future work.

  1. How did this new NLC compares to the traditional iron-oxide based NLCs?

Answer

As reported in literature (/Abbas et al., Superparamagnetic iron oxide-loaded lipid nanocarriers incorporated in thermosensitive in situ gel for magnetic brain targeting of clonazepam. Int. J. Pharm. 2018, 107, 2219-2117; Millart et al., Superparamagnetic lipid-based hybrid nanosystems for drug delivery. Expert Opinion on Drug Delivery, 2018, 15, 523-540), drug and iron oxide co-loaded lipid nanoparticles are superparamagnetic drug carriers entrapping magnetic nanoparticles that could be targeted to a specific tissue with the help of an external magnetic field. In the NLC investigated in this work, we co-loaded two drugs with synergic effect but none of them was supposed to show a targeting effect. Indeed, we designed NLC to improve IDE efficacy while a targeting effect was not the goal of our work.

Reviewer 2 Report

The authors have prepared a well-written manuscript that advances drug delivery using NLCs. They systematically investigate a variety of nanoformulations that reveal the excellent outcome of improved system properties along with potential improved efficacy. Clearly, a lot of hard work has gone into this effort. However, I do have one significant issue and a few other comments before I can recommend publication.

One thing I’m curious about is the colloidal nature of your samples. You suggest the systems are monodisperse. Your PDI values and size errors support that. However, your zeta values are close to the isoelectric point. I’ve read that colloidal suspensions are stable at more than positive 30 mV and less than negative 30 mV (Clogston, J.D., 2009, Measuring Zeta Potential of Nanoparticles. NCL Method PCC-2.). This makes me curious what the concentration of your nanoparticles were upon PCS and ZP measurement. Furthermore, how does that characterization concentration compare with the expected concentration of nanoparticles during the biomedical application? If those concentrations are very different, some discussion of this should be included.

Page 2 of 13, lines 93-94: I recommend your line should read: “Lipid nanoparticle samples were stored in airtight vials at room temperature and sheltered from the light until used.”

Page 3 of 13, lines 106-107: It would be interesting to know the concentration of the sample from which the grid was prepared for the purposes of replicating your images. In particular, novice researchers will value that detail.

Page 4 of 13, line 146: The subscripts in the equation are too long. I recommend abbreviating to make the equation portable.

Page 7 of 13, Figs. 3 and 4: These graphs were of such low resolution that I could not review them. The labels and scale text cannot be discerned at 100% magnification.

Page 8 of 13, Fig. 5: I could not review these bar charts. The text is illegible.

Author Response

Reviewer 2

The authors have prepared a well-written manuscript that advances drug delivery using NLCs. They systematically investigate a variety of nanoformulations that reveal the excellent outcome of improved system properties along with potential improved efficacy. Clearly, a lot of hard work has gone into this effort. However, I do have one significant issue and a few other comments before I can recommend publication.

One thing I’m curious about is the colloidal nature of your samples. You suggest the systems are monodisperse. Your PDI values and size errors support that. However, your zeta values are close to the isoelectric point. I’ve read that colloidal suspensions are stable at more than positive 30 mV and less than negative 30 mV (Clogston, J.D., 2009, Measuring Zeta Potential of Nanoparticles. NCL Method PCC-2.). This makes me curious what the concentration of your nanoparticles were upon PCS and ZP measurement. Furthermore, how does that characterization concentration compare with the expected concentration of nanoparticles during the biomedical application? If those concentrations are very different, some discussion of this should be included.

Answer

We would like to thank the reviewer for reviewing our manuscript and for his/her comments.

We agree with the reviewer that, as reported in literature, colloidal suspensions are expected to be stable at more than positive 30 mV and less than negative 30 mV. As the reviewer noticed, the colloidal suspensions we prepared showed zeta potential values close to the isoelectric point. This was the reason why we performed long-term stability studies that pointed out a good stability of lipid nanoparticles loading IDE percentages smaller than 1.5% w/w. In previous works (Montenegro et al. In vitro evaluation of idebenone-loaded solid lipid nanoparticles for drug delivery to the brain. Drug Dev. Ind. Pharm. 2011, 37, 737–746; Montenegro et al., In vitro evaluation on a model of blood brain barrier of idebenone-loaded solid lipid nanoparticles. J. Nanosci. Nanotechnol. 2012, 12, 330–337), we have already obtained lipid nanoparticles showing low zeta potential values but good stability. The good stability of these nanoparticles could be due to the mixture of surfactant and co-surfactant we used to prepare these colloidal suspensions. The presence of the same lipophilic chain (oleic acid) in the surfactant and co-surfactant structure could favor the interactions between surfactant and co-surfactant making the resulting film on the nanoparticle surface more flexible and resistant, thus preventing nanoparticle aggregation during storage, despite of their low zeta potential value. We have already suggested this explanation in a previous work.

As reported in the manuscript (page 4), samples were diluted 1:4 v/v with filtered deionized water prior to perform PCS measurements. The first time we prepared lipid nanoparticles using the phase inversion temperature method, we performed experiments to evaluate the effect of sample dilution on particle size measurements to find the optimal experimental conditions. We did not observe any significant change of particle size for undiluted samples and samples diluted up to 1:10. We chose the dilution 1:4 to obtain an optimal peak intensity. We added in the text the sample dilution (1:1 v/v) we used to obtain zeta potential values. As reported in the manuscript, we described the method we used in a previous paper. All stability tests were performed on undiluted samples. Therefore, characterization concentration is not expected to affect concentration of nanoparticles during the biomedical application.

Page 2 of 13, lines 93-94: I recommend your line should read: “Lipid nanoparticle samples were stored in airtight vials at room temperature and sheltered from the light until used.”

Answer

To comply with the reviewer’s comment, we amended the text as requested.

Page 3 of 13, lines 106-107: It would be interesting to know the concentration of the sample from which the grid was prepared for the purposes of replicating your images. In particular, novice researchers will value that detail.

Answer

We used undiluted sample. We added this information in the text (page 3 line 106-107).

Page 4 of 13, line 146: The subscripts in the equation are too long. I recommend abbreviating to make the equation portable.

Answer

To comply with the reviewer’s request, we abbreviated the subscripts in the equation. We amended the text as follows (page 4 line 146):

RI(%) = [ΔHaq. lipid np disp./(ΔHbulk material x conc sol. lip. phase)]x100

Where ΔHaq. lipid np disp. = ΔH of the aqueous lipid nanoparticle dispersion,  ΔHbulk material was calculated as cetyl palmitate enthalpy variation and conc sol. lip. phase was the concentration of the solid lipid phase.

Page 7 of 13, Figs. 3 and 4: These graphs were of such low resolution that I could not review them. The labels and scale text cannot be discerned at 100% magnification.

Answer

To comply with the reviewer’s request, we modified Fig. 3 and 4.

Page 8 of 13, Fig. 5: I could not review these bar charts. The text is illegible.

Answer

To comply with the reviewer’s request, we modified Fig. 5.
